# Structural insights into ribosomal rescue by Dom34 and Hbs1 at near-atomic resolution

Tarek Hilal[1], Hiroshi Yamamoto[1], Justus Loerke[1], Jörg Bürger[1,2], Thorsten Mielke[2] & Christian M.T. Spahn[1]

The surveillance of mRNA translation is imperative for homeostasis. Monitoring the integrity of the message is essential, as the translation of aberrant mRNAs leads to stalling of the translational machinery. During ribosomal rescue, arrested ribosomes are specifically recognized by the conserved eukaryotic proteins Dom34 and Hbs1, to initiate their recycling. Here we solve the structure of Dom34 and Hbs1 bound to a yeast ribosome programmed with a nonstop mRNA at 3.3 Å resolution using cryo-electron microscopy. The structure shows that Domain N of Dom34 is inserted into the upstream mRNA-binding groove via direct stacking interactions with conserved nucleotides of 18S rRNA. It senses the absence of mRNA at the A-site and part of the mRNA entry channel by direct competition. Thus, our analysis establishes the structural foundation for the recognition of aberrantly stalled 80S ribosomes by the Dom34•Hbs1•GTP complex during Dom34-mediated mRNA surveillance pathways.

[1] Institut für Medizinische Physik und Biophysik, Charité-Universitätsmedizin Berlin, Charitéplatz 1, 10117 Berlin, Germany. [2] Max-Planck Institut für Molekulare Genetik, Ihnestrasse 63-73, 14195 Berlin, Germany. Correspondence and requests for materials should be addressed to C.M.T.S. (email: christian.spahn@charite.de).

The ribosome is the macromolecular machine that synthe-sizes genetically encoded proteins in all domains of life. Protein synthesis is a highly regulated process that is coupled to a variety of quality control mechanisms[1]. Monitoring the integrity of mRNA is essential not only to minimize the production of aberrant, potentially harmful proteins, but also to rescue ribosomes stalled on defective messenger RNAs. To maintain the pool of translation competent complexes, ribosomes translating canonical mRNAs are usually recycled in tandem with the termination phase of protein synthesis as coupled processes in eukaryotes. The ability of the elongating ribosome to reach a canonical stop codon is crucial for initiating these processes. Aberrant mRNAs, where this ability is compromised, trigger an mRNA surveillance pathway[2–4]. This pathway is activated on mRNAs lacking a stop codon (nonstop mediated decay, NSD[5]) or on mRNAs with a strong secondary structure capable of stalling translation elongation (no-go mediated decay, NGD[6]). Furthermore, mRNAs containing rare codons, poly(A) stretches[7] or damaged mRNAs[8] elicit a quality-control response.

mRNA surveillance ultimately leads to exosomal degradation of the aberrant mRNAs and couples with the downstream ribosome quality-control pathway[9,10], which results in ubiquitination, extraction and degradation of the nascent-peptide chain. These steps are preceded by recognition and splitting of the stalled 80S ribosomes. Crucial for the recognition step are the conserved protein factors Dom34 (Pelota in mammals) and the translational eukaryotic elongation factor 1A (eEF1A)-like GTPase Hbs1 (refs 11,12). These are homologous and related in structure to the canonical translation termination factors eRF1 and eRF3, respectively. Dom34, similar to eRF1, can be structurally divided into the amino-terminal domain (N), a middle domain (M) and the carboxy-terminal domain (C), which appear highly similar between both homologues. Hbs1 consists of a unique N-terminal domain of ∼160 amino acids, followed by Domain G and the β-barrel domains 2 and 3, which are highly similar to eRF3 and eEF1a.

Similar to termination, where the stop codon is decoded by the eRF1•eRF3•GTP ternary complex, stalled ribosomes on non stop or no-go mRNAs are recognized by the Dom34•Hbs1•GTP ternary complex. Unlike eRF1, however, the C domain of Dom34 does not contain the GGQ motif crucial for nascent-chain hydrolysis[11,13]. Thus, following GTP hydrolysis and dissociation of Hbs1•GDP, Dom34 is not capable of hydrolysing the peptidyl-transfer RNA, which remains bound to the 60S subunit after subsequent splitting of the 80S ribosome by accommodated Dom34 in concert with the eukaryotic recycling factor Rli1/ABCE1 (ref. 14).

The structures of Dom34 (ref. 15), Hbs1 (ref. 16) and the Dom34•Hbs1•GMPPNP complex from *Schizosaccharomyces pombe*[17] and the archaeal aPelota•aEF1•GTP[18] have been solved by X-ray crystallography. Furthermore, cryo-electron microscopy (cryo-EM) at intermediate resolution has visualized Dom34•Hbs1•GMPPNP bound to a ribosomal yeast 80S no-go decay complex[19] with the ribosomal factor binding site in a position highly similar to the eRF1•eRF3•GMPPNP complex during canonical translation termination[20–22]. However, although the structural basis for eukaryotic stop codon recognition by eRF1 during canonical translation termination has been recently established by cryo-EM at near-atomic resolution[23], the molecular details how the Dom34•Hbs1•GTP complex recognizes stalled 80S ribosomes remain elusive.

Here we present a cryo-EM structure of the Dom34•Hbs1•GMPPNP complex bound to the yeast 80S ribosome programmed with a nonstop mRNA at 3.3 Å resolution. Our results show that Dom34 competes with the mRNA for binding at the decoding centre by directly stacking with the conserved nucleotides of 18S rRNA, thus revealing the molecular basis of the initial step of Dom34-mediated quality-control pathways.

## Results

**Reconstitution and overall confirmation of the nonstop ribosomal complex.** To mimic the situation of ribosomal complexes stalled on nonstop mRNAs, we used a minimal *in vitro* system. We first programmed purified 80S ribosomes from *Saccharomyces cerevisiae* with a short model mRNA and bound acetylated AcPhe-tRNA$^{Phe}$ from yeast to an unique UUC codon preceding two nucleotides from the 3′-end of the mRNA. Subsequently, we added the ternary complex of Dom34•Hbs1•GMPPNP and directly subjected the reaction to cryo-EM analysis. Multiparticle refinement[24] resulted in a cryo-EM reconstruction of the desired nonstop ribosomal complex (NsRC) with an overall resolution of 3.3 Å (Supplementary Figs 1 and 2, and Supplementary Table 1).

In terms of overall ribosomal conformation, the NsRC is related to the previously described post-translocational state (POST state)[25]. The ribosome is in the non-rotated conformation with a classical P/P-site bound tRNA (Fig. 1). Similar to the previous intermediate-resolution cryo-EM analysis of a stem-loop stalled NGD complex[19], Dom34 occupies the A-site region forming mainly contacts to the 40S subunit and the core of Hbs1 is bound in the GTPase-associated centre between the ribosomal 40S and 60S subunits. In addition, the N-terminal extension of Hbs1 is placed between uS3 of the 40S head and h16 of the 40S body (Fig. 1 and Supplementary Fig. 3). In contrast to the previous study of the NGD complex[19], we can clearly trace the mRNA path, which ends after the P-site codon. The two terminal nucleotides of our mRNA construct are fragmented (Fig. 1b,c). As the A-site is free of mRNA, our complex indeed represents a nonstop stalled ribosome that is derived *in vivo* from the translation of stop-codon-less mRNAs[26].

**Architecture of the ribosome bound Dom34•Hbs1 complex.** The eukaryotic rescue factors Dom34 and Hbs1 form a stable complex already in the absence of the ribosome[15,17]. Crystal structures of ribosome-free homologous factors in the presence[18] or absence[17] of a nucleotide suggest multiple interactions between Dom34 and Hbs1. In addition, in the present yeast NsRC, Dom34 and Hbs1 share a large interaction surface including two domains of Dom34 and three domains of Hbs1 (Fig. 2). Domain 3 of Hbs1 is enclosed by three α-helices of Dom34, in particular α7, α8 and α11 (Fig. 2a). Noticeably, the C-terminal end of α8 is inserted into a binding pocket of Hbs1, and especially Leu288, Leu291 and Asn292 of Dom34 undergo extensive interactions with Arg517, Pro518 and His558 of Hbs1. Furthermore, Glu361, Gln364 and Leu365 (helix α11) of Dom34 are involved in a hydrogen bond network with Gly522 and Pro521 of Hbs1. Besides being involved in intramolecular interactions within Dom34, the highly conserved Tyr300 also directly positions its hydroxyl group in bonding distance to Hbs1 (Supplementary Fig. 5). Mutational studies show a significant reduction in ribosome rescue efficiency[18], suggesting an important role of Tyr300 in the Dom34/Hbs1 complex. In contrast to the ribosome-free complex of Dom34 and Hbs1 (ref. 12), however, we cannot find a direct interaction between Glu169 (Asp 164 in *S. pombe*) and Gln487 (Arg480 in *S. pombe*). Instead, Arg144 and Ser145 of ribosomal protein uS12 are complexing an ion serving as a hub between the rescue factors (Supplementary Fig. 4a).

A remarkable feature of Dom34's middle domain is the long helix α3, which is protruding between domain 2 and G of Hbs1. Its tip is harbouring a flexible loop, which is not very well defined in our cryo-EM structure. However, numerous contacts to Hbs1

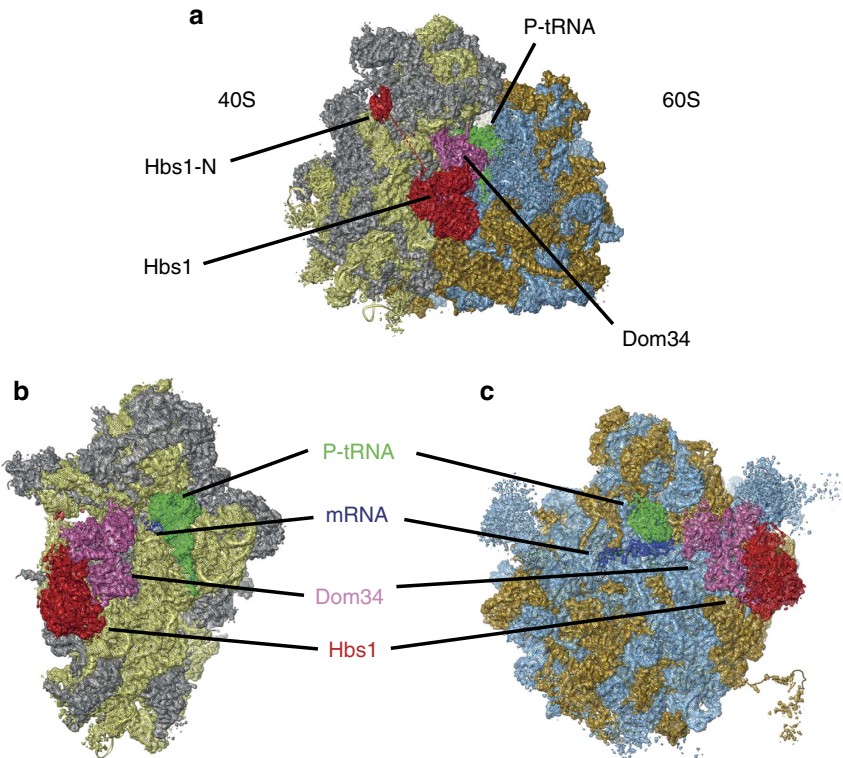

**Figure 1 | Electron density map of the *S. cerevisiae* ribosome stalled on nonstop mRNA.** (**a**) Overview of the 80S ribosome with bound ligands. 40S rRNA is coloured yellow, 40S proteins are grey, 60S rRNA is blue and 60S proteins orange. The N-terminal domain of Hbs1 (red) is located ~60 Å away from the core. (**b**) Isolated density of the 40S subunit with factors. The mRNA (blue) is ending after the P-tRNA. (**c**) 60S subunit with factors.

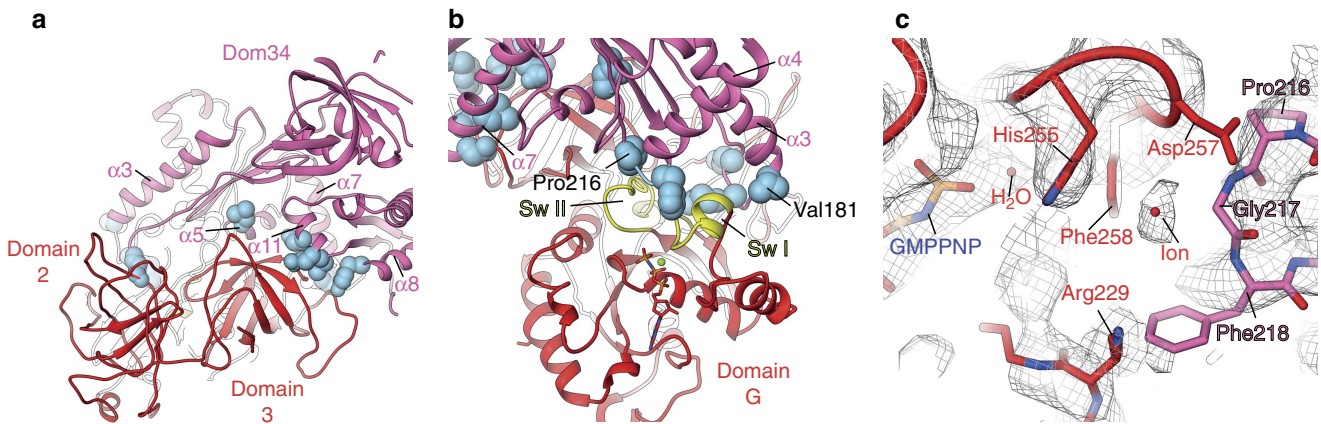

**Figure 2 | Interaction sites between Hbs1 and Dom34.** Dom34 (hot pink) and Hbs1 (red) interact at multiple sites. (**a**) Domain 3 of Hbs1 undergoes contacts with helices 5, 7, 8 and 11 of Dom34. Contact sites are highlighted as blue spheres. (**b**) Switch I and II of Hbs1 (yellow) are contacted directly by helices 3 and 4 of Dom34. Pro216 of the conserved PGF motif is close to switch II; Val181 is interacting with switch I. (**c**) The PGF motif is interacting with Arg229 of switch I and Asp257 of switch II. An ion is coordinated between the essential His255 of Hbs1 and Gly217 of Dom34.

are likely to be due to the tight packing of Dom34 against the GTPase factor. In the X-ray structure of the vacant complex[12] the G domain of Hbs1 is rotated away from Dom34 by 42°. The position of the Hbs1 G domain in our NsRC is similar to the archaeal aPelota•aEF1•GTP complex[18], however, facilitating additional contacts with Dom34. Residing just at the end of helix α3, Val181 interacts with Asn224 of the switch I region of Hbs1 (Fig. 2b). Directly situated in front of helix α4 is the PGF motif, a sequence motif highly conserved among Dom34 homologues from all species. Our structure shows close proximity of Pro216 and Gly217 to switch II of Hbs1, in particular Asp257

and Phe258 (Fig. 2b,c). Asp257 forms a hydrogen bridge to His250 of Dom34. We find an ion coordinated between the catalytic His255 of Hbs1 and P/G of the PGF motif that may be of importance for the positioning of His255. In addition to contacts with switch II of Hbs1, the PGF motif is interacting with switch I as well. Phe218 of Dom34 is in close proximity to the side chains of Glu225 and Arg229, giving rise to stacking interactions (Fig. 2c).

The G domain of Hbs1 is homologous to elongation factor EF-Tu/eEF1a, which delivers aminoacyl-tRNA to the ribosomal A-site in form of a ternary complex[25,27]. A comparison of the

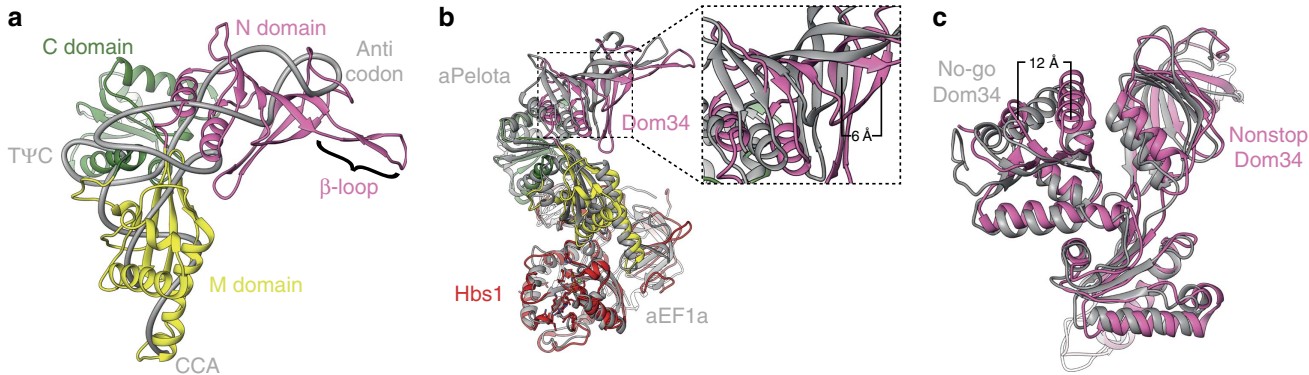

**Figure 3 | Comparison of Dom34 structures.** (**a**) Superposition of the atomic models for eukaryotic A/T tRNA (grey) during decoding and Dom34 on the nonstop-stalled ribosome. The N-terminal domain of Dom34 is coloured hot pink, domain M yellow and domain C green. Both models superpose to a large extent, except for the β-loop of Dom34. (**b**) Alignment of Dom34/Hbs1 and the archael aPelota/aEF1a complex on the G proteins. The N domains of Dom34 and aPelota are shifted by 6 Å against each other. (**c**) Structures of Dom34 generally overlap well during nonstop and no-go recognition; however, in particular, domain C shows significant conformational changes.

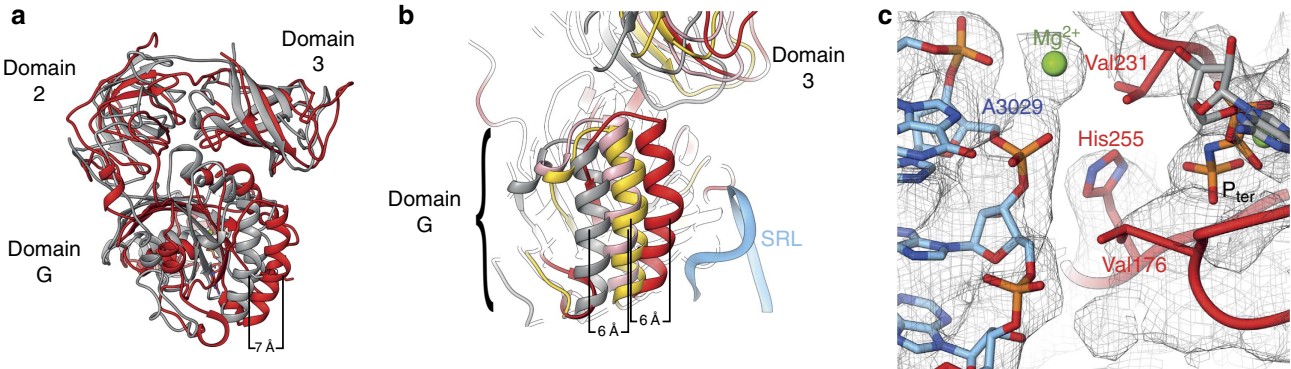

**Figure 4 | Activation of G-proteins in eukaryotes.** (**a**) Superposition of *S. cerevisiae* Hbs1 during nonstop recognition (red) and no-go decay (grey). Domains 2 and 3 overlap to a large extent, in contrast to domain G. (**b**) Superposition of the G domains during codon sampling (grey), codon recognition (yellow), no-go decay (pink) and nonstop ribosomal rescue (red). The SRL of the 60S subunit is depicted in blue. The α-helix harbouring Ser330 can be found in three different positions being closest to the SRL in the NsRC. (**c**) The conserved His255 is positioned between A3029 of the SRL (blue) and the terminal phosphate (orange) within Hbs1. The hydrophobic gate between Val176 and Val231 is open.

mammalian ternary complex from a cryo-EM reconstruction at subnanometre resolution with our rescue factors shows that Dom34 largely overlaps with the A/T tRNA in a common alignment of the GTPase factors Hbs1 and eEF1A, respectively (Fig. 3a). Especially at the anticodon stem loop and the acceptor stem, the N and M domains, respectively, of Dom34 appear as a nearly perfect mimic of the A/T tRNA. The loop between β-sheets 5 and 6 of the Dom34 N domain are thereby placed at the position of the tRNA anticodon. The mimicry of the acceptor stem extends to most of the single-stranded 3′-CCA-end of the tRNA, which overlays well with Helix α3 of Dom34. Somehow more divergent is the position of Dom34 domain C compared with the tRNA elbow region. Besides the high overall similarity in shape, the N domain of Dom34 has a striking extension compared with the A/T tRNA build by the noteworthy β-loop connecting β-sheets 3 and 4 within the N domain (Fig. 3a).

**Characteristics of nonstop targeted rescue factors.** Our present cryo-EM structure represents the first structure of a ribosome rescue intermediate on a well-defined nonstop mRNA substrate. Interestingly, compared with the no-go decay intermediate at subnanometre resolution[19], we can find some significant differences in the location of the rescue factors that cannot be attributed to differences in resolution only, but may indicate

differences in the functional state. After alignment of the respective ribosomal 60S subunits, domains 2 and 3 of Hbs1 match reasonably well between both intermediates, but the G domain in our nonstop intermediate is shifted closer towards the 60S subunit and the sarcin ricin loop (SRL) by ~6–7 Å (Fig. 4a). Movement of a translational GTPase towards the SRL is reminiscent to mammalian decoding, where we previously described movement of the G domain of eEF1A towards the SRL from the codon sampling state to the codon recognition state of ternary complex[25].

To further assess the position of the G domain, we included sampling and recognition states of eEF1A during mRNA decoding[25] into our analysis. Superposition of the four structures reveals a trajectory of G domain movement approaching the 60S subunit with three distinct positions as exemplified by the α-helix harbouring Ser330 (Fig. 4b). In the codon sampling state, the G domain of eEF1A is not contacting the SRL, but a movement of ~6 Å to the codon recognition state brings the GTPase in contacting distance to the SRL. The position of Hbs1 in the no-go decay complex[19] matches the codon recognition state of eEF1A[25] well, differing only by ~1 Å. However, eEF1A in detail has a slightly different conformation undergoing additional contacts to the ribosome.

Remarkably, in the present nonstop complex Hbs1 is in a third position, bringing Ser330 additional ~6 Å closer to the 60S

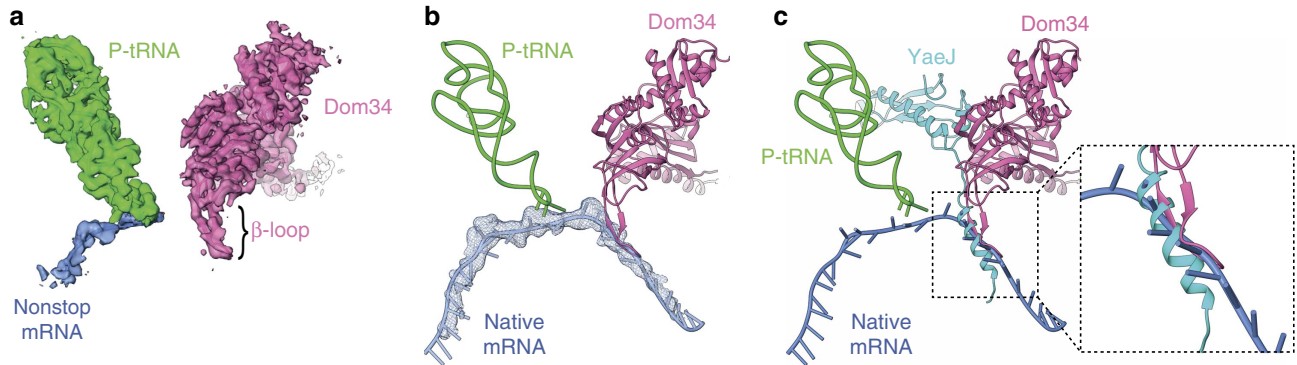

**Figure 5 | A unique β-loop of Dom34 senses mRNA. (a)** Segmented experimental electron density maps for P-tRNA (green), mRNA (blue) and Dom34 (hot pink). The nonstop mRNA is directly interacting with the anticodon of the P-tRNA. No additional mRNA density can be assigned emerging the P-site, indicating that the ribosome is positioned on the 3′ end of the mRNA. Dom34 is bound in the A-site, protruding its unique β-loop into the mRNA channel. **(b)** Superposition of atomic models for P-tRNA and Dom34 with mRNA from human polysomes[31] (blue). **(c)** Superposition of Dom34 (hot pink) in the yeast NsRC with the bacterial rescue factor YaeJ (cyan). The β-loop of Dom34 is replaced by an α-helix in YaeJ. Both proteins are unrelated but sense absence of mRNA in the A-site.

subunit (Fig. 4b). Assuming that activation of translational GTPases is mediated by tight interactions with the SRL[28], it appears that our structure closely resembles the GTPase activated state of Hbs1 during ribosomal rescue. Comparison with the structure of isolated Hbs1 in complex with GDP[16] further supports this idea, as switches I and II of the G domain are ordered and remodelled in our NsRC, presumably due to interactions with the SRL. The catalytic His255 (His85 in EF-Tu) is in a triangular position between the γ-phosphate and the universally conserved A3029 of the SRL (A2662 in *Escherichia coli*) having a distance of ~4 Å to both (Fig. 4c). This, as well as the open hydrophobic gate comprising Val176 and Val231 (Val20 and Ile60 in EF-Tu), strongly resemble the architecture presented for activated EF-Tu[28]. The evolutionary conservation of translational GTPases is further highlighted by comparison of Hbs1 in the NsRC with 80S-bound eEF2[29]. The core of the G domains of both enzyme subfamilies superpose well; however, positioning of the remaining domains is distinct.

Looking at the A-site binding factor Dom34, we can find a reasonable superposition of domains N and M between our NsRC and the NGD intermediate. In contrast, Dom34 domain C of both structures shows conformational changes. Helix α10 of Dom34 is rotated towards the P-site in our map, resulting in a distance of ~12 Å between Asn345 at the end of α10 in both structures (Fig. 3c). This may indicate conformational flexibility of Dom34 even in the ribosome-bound state.

Some conformational freedom of Dom34 can be also seen from a comparison of the present NsRC with the crystal structure of the homologous complex from archaea including aPelota, aEF1a and GTP[18]. Overall, the structures of yeast Hbs1 and aEF1a match very well. In addition, the Dom34 M domains, especially helix α3 and the Sm-fold including β sheets 8–12 superimpose nicely. However, we find differences for the same helices of domain C of Dom34 that were different between the nonstop and the no-go intermediate. Moreover, domain N is shifted as a rigid body by ~6 Å (measured at Ser106 of β6) to approach the 40S subunit (Fig. 3b). The apparent high degree of conformational flexibility of Dom34 explains the need for an unstructured linker between helix α2 of domain N and sheet β8 of domain C. It is tempting to speculate that this flexibility of Dom34 is needed for sampling a variety of ribosomal complexes.

**Structural details of nonstop stalled ribosome recognition.** The middle domain of Dom34 is tightly packed against Hbs1 and not involved in ribosomal binding. In contrast to its paralogue eRF1,

domain M of Dom34 does not contain a GGQ motif; hence, no catalytic activity is known. Besides undergoing multiple interactions with Hbs1, the C domain of Dom34 anchors it on the 60S subunit by forming stacking interactions between Tyr374 and G1242 of Helix 42 (H42) of the 25S rRNA (Supplementary Fig. 4b). The evolutionary conservation of aromatic residues from yeast to man pinpoints an important functional interaction site at this position. Dom34 is interacting with the 40S subunit mainly via its N-terminal domain. The N-terminal β-sheet of Dom34 is involved in binding of the 40S head. In the crystal structure of yeast Dom34 (ref. 15), the loop between β1 and β2 was disordered. We can see an ordering of this flexible region in our map, most probably due to ribosomal contacts.

A key question underlying our study was how, exclusively, aberrantly stalled ribosomes are recognized by Dom34/Hbs1. Biochemical studies emphasize codon independence[11], thus suggesting a recognition mechanism distinct from stop codon recognition by eRF1 (ref. 30). Besides being overall very similar to eRF1 (ref. 20), the N-terminal domains of both homologous proteins differ markedly. Notably, the extension of Dom34 connecting β-sheets 3 and 4 within the N domain expands the shape of the N domain compared with the tRNA anticodon stem loop (Fig. 3a) and eRF1. This loop (Thr48–Thr60), which was disordered in the crystal structure of Dom34 (ref. 15), folds as a β-loop and deeply penetrates into the mRNA channel (Fig. 5a). We note that this loop contains preferably large charged side chains, especially Lys, Asp and Glu, thus maximizing the spatial extent into the mRNA channel. Superposition with mRNA from native translating polysomes[31] (Fig. 5b) suggests that this β-loop mimics mRNA from position +5 (the second nucleotide of the A-site codon) until position +9. This suggests a function as a mechanical sensor, disfavouring binding to actively translating ribosomes carrying mRNA in the A-site and the upstream part of the mRNA entry channel.

Remarkably, the bacterial nonstop mRNA rescue factor YaeJ uses a similar recognition strategy. In contrast to the β-loop of Dom34 however, YaeJ inserts an α-helical element into the unoccupied mRNA channel[32] (Fig. 5c). In both domains of life, the essentiality of the respective free-mRNA-channel-sensing element is highlighted by mutational studies, where deletion of the corresponding elements impaired functions during ribosomal rescue[18,26,33]. As Dom34 and YaeJ are not related to each other, the strategy of mechanical sensing to identify ribosomes stalled on nonstop mRNAs has been developed during evolution at least twice.

Besides the function as a mechanical sensor in the mRNA channel, domain N of Dom34 interacts with the decoding centre

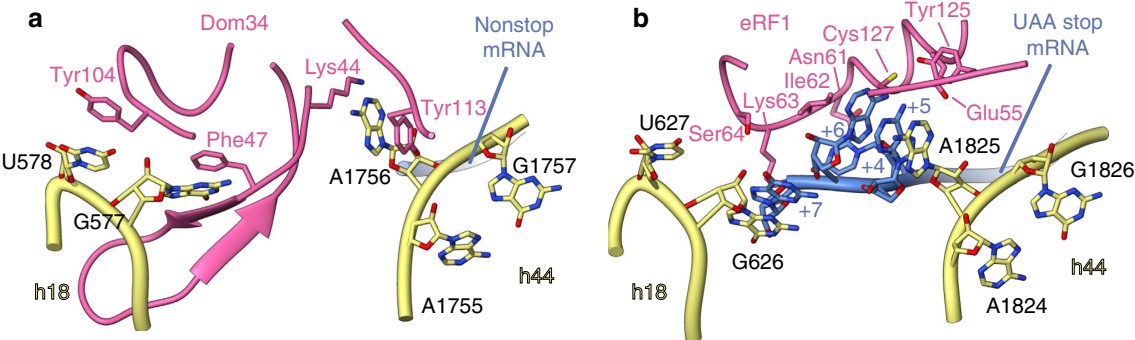

**Figure 6 | Structure of the ribosomal decoding centre during recognition of nonstop arrested ribosomes and canonical termination.** (**a**) Dom34 (hot pink) interacts directly with the ribosomal decoding centre of the 40S subunit (yellow). A1756 of the 18S rRNA (h44) is flipped out and stacks between Lys44 and Tyr113. G577 and U578 of h18 are stacking with Phe47 and Tyr104, respectively. (**b**) During canonical termination (ref. 30), the stop codon of the mRNA (blue) induces a similar configuration of the decoding centre. eRF1 (hot pink) mainly interacts with the mRNA, which replaces Dom34 and mediates most of the ribosomal contacts in the decoding centre.

of the 40S subunit at positions usually occupied by mRNA (Fig. 6a). A1756 (A1493 in *E. Coli* and A1825 in higher eukaryotes) of h44 was shown to flip out during eukaryotic termination due to stacking interactions with the second nucleotide of the stop codon[30] (Fig. 6b). Strikingly, Dom34 adopts the role of the mRNA in promotion of base flipping by embedding A1756 between Tyr113 and Lys44 of domain N. In addition, Dom34 interacts with h18, another integral part of the decoding centre. G577 (G530 in *E. Coli* and G626 in higher eukaryotes), usually stacking with the $+6$ nucleotide of the mRNA during decoding or $+7$ during eukaryotic stop codon recognition, is stacking with Phe47 of the β-loop of Dom34. Furthermore, we can find U578 involved in stacking interactions with Tyr104 of Dom34. This seems to be unique for recognition of aberrantly stalled ribosomes, as no similar interaction has been described for canonical decoding.

Taken together, domain N of Dom34 occupies the path of the mRNA from position $+5$ (the middle of the A-site codon) till $+9$ and competes with the mRNA downstream of the P-site codon. Apparently, the absence of mRNA in this ribosomal region is a critical determinant for the recognition of stalled ribosomes.

**The role of Hbs1 in recognizing stalled ribosomes.** Our cryo-EM reconstruction of the ribosomal 80S•Dom34•Hbs1• GMPPNP complex shows additional density at the back of the 40S subunit close to the mRNA entry channel placed between uS3 of the 40S head and h16 of the 40S body (Fig. 1a and Supplementary Fig. 3a). Following the path of mRNA in cryo-EM maps of programmed 80S ribosomes, for example our previous Cricket paralysis virus IRES (CrPV-IRES) bound ribosome[20], suggests that this position can be occupied by a 3′-part of the mRNA (positions $+14$ onwards; Supplementary Fig. 3b). As we used a truncated mRNA having only two nucleotides downstream of the P-site, we can exclude that the density in our map is representing mRNA. Based on the α-helical density, we can place a part of the N-terminal domain of Hbs1, in overall agreement with a previous study of the NGD complex[19]. The visible part of the N-terminal domain of Hbs1 consists of three densely packed α-helices, connected by short linkers. Binding of Hbs1 causes a displacement of the tip of h16 by 10 Å, widening the space between h16 and uS3 (Supplementary Fig. 3a). The linker region of Hbs1 (aa 78–140) appears disordered, as we could not identify connecting density between the N-terminal part and the core of Hbs1. Thus, the N-terminal domain of Hbs1 could act as sensor too, to support Dom34 in monitoring the absence of mRNA.

## Discussion

Our cryo-EM reconstruction of the yeast ribosomal 80S•Dom34•Hbs1•GMPPNP complex at 3.3 Å resolution provides the structural basis for the first phase of nonstop-mediated mRNA decay, with detailed insights into the molecular mechanism recognizing aberrantly stalled ribosomes. Domain N of rescue factor Dom34 binds in the ribosomal A-site and recognizes a particular conformation of the decoding centre, with a flipped out A1756 (A1493 in *E. coli*) and a flipped in, base-paired A1755 (A1492 in *E. coli*). The universally conserved G577, U578 and A1756 of the decoding centre undergo aromatic stacking interactions with specific amino acids of Dom34. A unique basic loop of Dom34 mimics part of the A-site codon and the upstream part of mRNA, thereby sensing the absence of mRNA by direct competition. Interestingly, mutational studies[18,26] emphasize the importance of this element for ribosomal rescue, showing severe reduction in non stop decay efficiency when deleted. Moreover, when co-expressed with native protein, mutant Dom34 (Dom34*) exhibited a dominant negative effect over wild type. As our findings suggest a crucial role of domain N for substrate recognition, binding of the Dom34*•Hbs1•GTP ternary complex to stalled ribosomes may be impaired. In this regard, Hbs1 is trapped in an unproductive ternary complex with mutant protein, reducing the cellular levels of available Hbs1 for complex formation with wild-type protein.

The ternary complex of Dom34•Hbs1•GTP has a dual sensor for the absence of mRNA: Dom34 probes A-site occupancy, whereas part of the N-terminal domain of Hbs1 probes upstream of the mRNA entry channel on the back of the 40S subunit. In addition, the tight interaction between Hbs1 and Dom34 maintains Dom34 in the inactive conformation, where Hbs1 clasps domain M of Dom34. Hbs1 itself has two functions: it holds Dom34 in an inactive state and impedes binding to translating ribosomes by sampling of the mRNA entry site. Dom34 works codon independently in promoting ribosomal splitting by Rli1/ABCE1. This reactivity could be obtained *in vitro* even in the absence of Hbs1; however, addition of Hbs1 increased the splitting rate 2.5-fold[12,34]. Hbs1 may improve binding of Dom34 to stalled ribosomes, but also may provide a means of tight regulation—in the presence of Hbs1, accommodation of Dom34 into its active conformation requires triggering of the GTPase activity of Hbs1. The delicate interplay between proper binding of Dom34 into the ribosomal decoding center and activation of GTP hydrolysis on Hbs1 may thus help to avoid accidental ribosomal splitting.

As Dom34 is not only involved in clearance of nonstop stalled ribosomes, but also during secondary-structure-induced NGD,

for example, the question arises how Dom34 can recognize such alternative situations. Biochemical experiments demonstrated an influence of 3′ mRNA length beyond the stall site on the ribosome splitting activity of Dom34 (Pelota in mammals), Hbs1 and Rli1 (ABCE1 in mammals)[12,34]. Increasing mRNA length downstream of the P-site was shown to be decisive for splitting activity, ultimately leading to complete abolishment. This effect could be reversed by incubation with RelE, a bacterial enzyme that cleaves mRNA in the A-site, clearly underlining the need for an accessible mRNA groove at the A-site[12] and further supporting the idea of a steric sensor. It is tempting to speculate that Dom34 may be able to dislocate shorter 3′-sequences of mRNA downstream of the P-site codon. Indeed, we cannot resolve nucleotides +4 and +5 of our model mRNA, indicating disorder. In the case of longer mRNAs, such as in NGD, endonucleolytic cleavage of the mRNA may be a prerequisite for Dom34 binding.

Endonucleolytic cleavage of the mRNA during NGD was shown in vivo[6] and is commonly accepted[3]; however, no endonuclease has been identified thus far. There have been speculations about the endonucleolytic activity of Dom34; however, the results of experiments with purified factors were ambiguous[13,35]. Mutational studies suggest that α-Helix 1, packed tightly inside the N-domain, may contain nuclease activity. In our structure of the ribosome-bound factor, this helix is far away from the mRNA channel, making a nucleolytic cleavage in this conformation unlikely. Nevertheless, cleavage by Dom34 in a different conformation, by free Dom34 in the cell or involvement of another, yet unidentified factor cannot be ruled out.

Interestingly, Becker et al.[19] noted the absence of density for the P-site codon in their intermediate resolution cryo-EM map of Dom34●Hbs1●GMPPNP bound to a ribosomal yeast 80S NGD complex and proposed that Dom34 binding may destabilize mRNA and disrupts codon–anticodon interaction in the P-site. However, in the present structure mimicking the NSD situation, we find the P-site codon well-ordered and resolved. We conclude that it is not Dom34 binding per se that destabilizes mRNA, but the expulsion of the downstream region of mRNA from its canonical path. Thus, endonucleolytic cleavage of the mRNA during NGD may be required for the expulsion of mRNA before NGD and NSD converge on Dom34●Hbs1●GTP binding to a similar ribosomal intermediate.

## Methods

**Preparation of factors.** Expression plasmids for His-tagged Dom34 and Hbs1 (refs 15,16) were transformed into E. coli BL21 (DE3) cells with antibiotic selection. Induction of protein expression was achieved with 0.5 mM isopropyl-β-D-thiogalactoside at $A_{600} = 0.6$–0.8. Cells were collected after 16 h expression at 18 °C and lysed by ultrasonication in lysis buffer (20 mM Tris-HCl pH 7.5, 250 mM NaCl, 5 mM β-mercaptoethanol, 1 × Complete protease inhibitor cocktail (Roche) and 25 mM imidazole). After clarification by centrifugation, lysates were applied to Nickel-NTA affinity purification followed by size-exclusion chromatography.

**Preparation of ribosomes.** Ribosomes were purified as described previously[36] with minor modifications. Briefly, mid-log phase S. cerevisiae cells were collected ($A_{600} \sim 0.8$) and lysed with a microfluidizer. After differential centrifugation (S10, S30 and S100), crude 80S ribosomes were dissociated with 1 mM puromycin and 1 mM GTP in high-salt buffer (50 mM Hepes pH 7.5, 3 mM $MgCl_2$, 500 mM KCl and 2 mM dithiothreitol (DTT)). After zonal centrifugation, ribosomal subunits were pooled individually, pelleted and reassociated in low-salt buffer (20 mM Hepes pH 7.5, 5 mM $MgCl_2$, 100 mM KCl and 2 mM DTT) during another zonal run. Final 80S ribosomes are free of protein factors, tRNA and mRNA.

**Sample preparation.** Reassociated 80S ribosomes (10 pmol) were incubated with 20 pmol acetylated Phe-tRNA$^{Phe}$ and 100 pmol SDF-mRNA (5′-GGCAAGGAG GUAAAAUUCUA-3′, IBA Germany) in binding buffer (30 mM Hepes pH 7.5, 5 mM MgOAc, 10 mM $NH_4OAc$, 1.2 mM spermidine, 5 mM β-mercaptoethanol, 1 mM DTT, 40 mM KOAc and 9% Glycerol). Separately, 50 pmol Dom34 and

50 pmol Hbs1 were incubated in binding buffer supplemented with 1 mM GMPPNP and additional 1 mM MgOAc. After 20 min at 30 °C, both reactions were combined and further incubated for 15 min at 30 °C.

**Electron microscopy.** An appropriately diluted aliquot of the sample (3.5 μl) was applied to glow-discharged Quantifoil R2/1 holey carbon films (Quantifoil Micro Tools GmbH), blotted for 2–4 s and flash frozen in liquid ethane using a Vitrobot (FEI).

Images were recorded fully automated using the LEGINON[37] software on a FEI Tecnai G² Polara microscope equipped with a Gatan K2 Summit direct electron detector in super resolution mode at 300 kV. The nominal magnification was 80,645, yielding a pixel size of 0.62 Å. Twenty-five movie frames were recorded during a total exposure time of 5 s with a corresponding dose of $\sim 25\,e^{-}\,Å^{-2}$. Defocus values varied between − 0.5 and − 4.5 μm.

**Image processing.** Movie stacks were corrected for beam-induced motion using all 25 image frames with MotionCorr[38]. Defocus estimation was done with CTFFIND4 (ref. 39). A total of 520,612 ribosomal particles were selected automatically using Signature[40] from 4,797 micrographs. Particle images were further processed with the SPIDER software package[41]. Briefly, starting with a low-resolution empty yeast ribosome as initial reference, we iteratively classified ribosomal particles using our multi-reference refinement approach[24] until homogeneity (Supplementary Fig.1). We obtained a stable population containing 82,227 particles that were subjected to particle polishing[42]. We attempted to further improve the reconstruction and used 2D classification within RELION[43]. Finally, 73,391 particles were assigned to the desired NsRC, additional three-dimensional classification confirmed homogeneity. The final refinement was done with FREALIGN[44]. However, the later steps did not lead to a significant further improvement of the map quality. Average resolutions were calculated using the FSC 0.143 criterion (semi-independent) and local resolutions were calculated with Resmap[45] (Supplementary Fig.2).

**Model building.** Initially, the crystal structures of S. cerevisiae 80S (4V7R)[46], Dom34 (2VGN)[15], Hbs1 (3P27)[16] and yeast Phe-tRNA (1EHZ)[47] were docked into our cryoEM map. The N-terminal domain of Hbs1 was homology modelled based on an NMR structure of BAB28515 (1UFZ).

In general, crystal structures were first rigid-body docked domain wise with Chimera[48] and reconnected in Coot[49]. Poorly fitting parts were then manually corrected and RNA was corrected semi-automatically with Rcrane[50] before refinement.

**Model refinement.** We applied real-space refinement[51] of the generated models in PHENIX[52] using secondary structure restraints derived from the initially used reference models with an optimized weight of 1. Overfitting was monitored by cross-correlating the refined model with half-sets of the data as described previously[53]. The final model was evaluated using the Molprobity algorithms within PHENIX.

**Data availability.** Atomic coordinates for the reported model have been deposited with the Protein Data Bank under accession code 5M1J. The electron density map has been deposited with the EMDB under accession code EMD-4140. The data that support the findings of this study are available from the corresponding author upon request.

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

## Acknowledgements

We thank K. Yamamoto for assistance with ribosome preparation and supply of the tRNA, B. Schroeer for assistance with the zonal centrifugation and Marc Graille for kindly sharing expression plasmids of Dom34 and Hbs1 with us. We thank M.L. Kraushar for comments on the manuscript. This work was supported by Grants from Deutsche Forschungsgemeinschaft DFG (SFB 740 to C.M.T.S. and T.M. and FOR1805 to C.M.T.S.) and HFSP grant RGP0062/2012 to C.M.T.S. We acknowledge the North-German Supercomputing Alliance (HLRN) for providing high-performance computing resources that have contributed to the research results reported in this paper.

## Author contributions

T.H. purified protein factors, ribosomes and prepared the sample together with H.Y. J.B. and T.M. performed cryo-EM and data collection. J.L. made the movie correction and particle polishing. T.H. calculated the cryo-EM reconstruction, built atomic models and interpreted the structure. T.H. prepared figures. T.H. and C.M.T.S. wrote the manuscript. All authors discussed and commented on the manuscript.

## Additional information

**Competing financial interests**: The authors declare no competing financial interests.

**Reprints and permission** information is available online at http://npg.nature.com/ reprintsandpermissions/

**How to cite this article**: Hilal, T. et al. Structural insights into ribosomal rescue by Dom34 and Hbs1 at near-atomic resolution. Nat. Commun. 7, 13521 doi: 10.1038/ncomms13521 (2016).

**Publisher's note**: 

