## [Peer Review File · Nature Communications]

Reviewers' Comments:

Reviewer #1 (Remarks to the Author):

The manuscript by Hilal et al is an outstanding and timely contribution to our emerging understanding of co-translational quality control in eukaryotes. Starting with a clever in vitro approach to reconstituting 80S stalling on a non-stop message, the authors employed state of the art cryo-EM imaging and analysis to visualize directly how Dom34 detects the empty A-site of stalled ribosomes. The chemistry of stalled ribosome recognition and splitting has been studied before, but never at this resolution. The authors make a compelling case that their updated views of the interactions between Dom34 and Hbs1, between Hbs1 and the ribosome, and between Dom34 and the A-site are of critical functional significance. I am especially excited about the new views of GTPase activation and the conformational changes that evolve from initial sampling to activation of the GTPase; and the role of the N-terminus of Hbs1 in probing the mRNA entry channel. Finally, the manuscript is well written, the figures are beautiful and easy to understand. I hope this report is published without delay.

Reviewer #2 (Remarks to the Author):

The conserved eukaryotic proteins Dom34 and Hbs1 specifically recognize ribosomes stalling at the 3' end of mRNA and facilitate its dissociation into subunits in concert with ABCE1. The structures of Dom34, Hbs1 as well as the Dom34•Hbs1•GMPPNP complex from *S. pombe* and the archaeal aPelota•aEF1•GTP have been solved by X-ray crystallography. Cryo-electron microscopy (cryo-EM) at intermediate resolution has visualized Dom34•Hbs1•GMPPNP bound to ribosome stalling in a position highly similar to the eRF1•eRF3•GMPPNP complex during canonical translation termination. However, the loop between β -sheets 5 and 6 of the Dom34 N-domain (Thr48-Thr60) are thereby placed at the position of the tRNA anticodon, was disordered in the crystal structure of Dom34. Therefore, the structural basis for Dom34-Hbs1 preferentially recognizes stalled ribosome with an empty A-site remains to be resolved. In this study, the structure of Dom34-Hbs1 complex bound to a yeast ribosome stalled at the 3' end of a nonstop mRNA at 3.3Å resolution using cryo-electron microscopy. I feel that the paper will influence thinking in the field, and would recommend this manuscript for publication, as there are major points that remain novel to this study. Firstly, Domain N of Dom34 is inserted into the upstream mRNA binding groove via direct stacking interactions with conserved nucleotides of 18S rRNA. It senses the absence of mRNA at the A-site and part of the mRNA entry channel by direct competition between Dom34 and mRNA. The β -loop (Thr48-Thr60) mimics mRNA from position +5 (the 2nd nucleotide of the A-site codon) until position +9. This presented results

provide the novel structural basis for a function of Dom34 as a mechanical sensor, disfavoring binding to ribosome stalled at the 3' end of nonstop mRNA with an empty A-site, but not to actively translating ribosomes carrying mRNA in the A-site but. Secondly, Hbs1 is in a third position in the present nonstop complex, bringing Ser330 additional $\sim 6 \text{ \AA}$ closer to the 60S subunit, and the structure closely resembles the GTPase activated state of Hbs1 for dissociation of stalled ribosome into subunits, a triggering step for NSD quality control. There are several points concerning the activity of Dom34 F47A and Dom34- Δ (F47-T60) mutant proteins in the recognition and dissociation of stalled ribosome at the 3' end of nonstop mRNA with an empty A-site that should be addressed prior to publication.

Major point

The yeast Dom34-F47A and Dom34- Δ (F47-T60) mutant protein has almost no activity for the production of nonstop protein derived from nonstop mRNA (Tsuboi et al., 2012). However, Dom34-F47A and Dom34- Δ (F47-T60) proteins exert dominant-negative effects on the activity of wild-type Dom34 (Tsuboi et al., 2012). Therefore, Dom34-F47A and Dom34- Δ (F47-T60) mutant protein may be defective in the subunit dissociation after the recognition of stalled ribosome. The structure of the ribosomal decoding center during recognition of nonstop arrested ribosomes resolved in this study demonstrated that G577 of h18 are stacking with Phe47 and β -loop (Thr48-Thr60) mimics mRNA from position +5 until position +9. Based on this structure, the authors may discuss the function of Phe47 and β -loop (Thr48-Thr60) in the recognition of an empty A-site as well as the subunit dissociation that depends on the interaction of Dom34 with ABCE1.

Point-by-point response to the referees

Reviewer #1 (Remarks to the Author):

The manuscript by Hilal et al is an outstanding and timely contribution to our emerging understanding of co-translational quality control in eukaryotes. Starting with a clever in vitro approach to reconstituting 80S stalling on a non-stop message, the authors employed state of the art cryo-EM imaging and analysis to visualize directly how Dom34 detects the empty A-site of stalled ribosomes. The chemistry of stalled ribosome recognition and splitting has been studied before, but never at this resolution. The authors make a compelling case that their updated views of the interactions between Dom34 and Hbs1, between Hbs1 and the ribosome, and between Dom34 and the A-site are of critical functional significance. I am especially excited about the new views of GTPase activation and the conformational changes that evolve from initial sampling to activation of the GTPase; and the role of the N-terminus of Hbs1 in probing the mRNA entry channel. Finally, the manuscript is well written, the figures are beautiful and easy to understand. I hope this report is published without delay.

We appreciate the comments of the reviewer and are happy about the positive feedback.

Reviewer #2 (Remarks to the Author):

*The conserved eukaryotic proteins Dom34 and Hbs1 specifically recognize ribosomes stalling at the 3' end of mRNA and facilitate its dissociation into subunits in concert with ABCE1. The structures of Dom34, Hbs1 as well as the Dom34•Hbs1•GMPPNP complex from *S. pombe* and the archaeal aPelota•aEF1•GTP have been solved by X-ray crystallography. Cryo-electron microscopy (cryo-EM) at intermediate resolution has visualized Dom34•Hbs1•GMPPNP bound to ribosome stalling in a position highly similar to the eRF1•eRF3•GMPPNP complex during canonical translation termination. However, the loop between β -sheets 5 and 6 of the Dom34 N-domain (Thr48-Thr60) are thereby placed at the position of the tRNA anticodon, was disordered in the crystal structure of Dom34. Therefore, the structural basis for Dom34-Hbs1 preferentially recognizes stalled ribosome with an empty A-site remains to be resolved.*

We thank the reviewer for the nice introduction.

In this study, the structure of Dom34-Hbs1 complex bound to a yeast ribosome stalled at the 3' end of a nonstop mRNA at 3.3Å resolution using cryo-electron microscopy. I feel that the paper will influence thinking in the field, and would recommend this manuscript for publication, as there are major points that remain novel to this study. Firstly, Domain N of Dom34 is inserted into the upstream mRNA binding groove via direct stacking interactions with conserved nucleotides of 18S rRNA. It senses the absence of mRNA at the A-site and part of the mRNA entry channel by direct competition

between Dom34 and mRNA. The β -loop (Thr48-Thr60) mimics mRNA from position +5 (the 2nd nucleotide of the A-site codon) until position +9. This presented results provide the novel structural basis for a function of Dom34 as a mechanical sensor, disfavoring binding to ribosome stalled at the 3' end of nonstop mRNA with an empty A-site, but not to actively translating ribosomes carrying mRNA in the A-site but. Secondly, Hbs1 is in a third position in the present nonstop complex, bringing Ser330 additional $\sim 6 \text{ \AA}$ closer to the 60S subunit, and the structure closely resembles the GTPase activated state of Hbs1 for dissociation of stalled ribosome into subunits, a triggering step for NSD quality control.

We are grateful for the concise summary.

There are several points concerning the activity of Dom34 F47A and Dom34- Δ (F47-T60) mutant proteins in the recognition and dissociation of stalled ribosome at the 3' end of nonstop mRNA with an empty A-site that should be addressed prior to publication.

Major point

The yeast Dom34-F47A and Dom34- Δ (F47-T60) mutant protein has almost no activity for the production of nonstop protein derived from nonstop mRNA (Tsuboi et al., 2012). However, Dom34-F47A and Dom34- Δ (F47-T60) proteins exert dominant-negative effects on the activity of wild-type Dom34 (Tsuboi et al., 2012). Therefore, Dom34-F47A and Dom34- Δ (F47-T60) mutant protein may be defective in the subunit dissociation after the recognition of stalled ribosome. The structure of the ribosomal decoding center during recognition of nonstop arrested ribosomes resolved in this study demonstrated that G577 of h18 are stacking with Phe47 and β -loop (Thr48-Thr60) mimics mRNA from position +5 until position +9. Based on this structure, the authors may discuss the function of Phe47 and β -loop (Thr48-Thr60) in the recognition of an empty A-site as well as the subunit dissociation that depends on the interaction of Dom34 with ABCE1.

One of our major findings is that the β -loop within Dom34 (Thr48-Thr60) acts as sterical sensor for nonstop recognition. In addition Phe47 is directly involved in binding of the ribosomal decoding center by stacking with G577. The reviewer nicely reflects the importance of this domain further supporting it with mutational studies by Tsuboi et al. (2012). The loss of activity during nonstop-decay confirms that the above mentioned elements are crucial for ribosomal rescue.

According to our results we suggest that binding of the mutant Dom34•Hbs1 complex to stalled ribosomes is impaired. As we now discuss on page 13, the observed dominant negative effect of mutant Dom34 can be explained by formation of non-productive ternary complexes with Hbs1 and GTP that cannot stably bind to arrested ribosomes, thereby sequestering Hbs1. The importance of a certain balance between Dom34 and Hbs1 in the cell is further exemplified by a non-functional Hbs1-mutant (Hbs1H255A) which would allow binding of the ternary complex to ribosomes but prevent downstream reactions.